# Revisiting Large Language Models as Zero-shot Relation Extractors

**Guozheng Li**$^{\diamond}$ and **Peng Wang**$^{\diamond\clubsuit(\boxtimes)}$ and **Wenjun Ke**$^{\diamond\clubsuit}$

$^{\diamond}$ School of Computer Science and Engineering, Southeast University, China
$^{\clubsuit}$ Key Laboratory of New Generation Artificial Intelligence Technology and Its
Interdisciplinary Applications (Southeast University), Ministry of Education, China
{liguozheng, pwang, kewenjun}@seu.edu.cn

## Abstract

Relation extraction (RE) consistently involves a certain degree of labeled or unlabeled data even if under zero-shot setting. Recent studies have shown that large language models (LLMs) transfer well to new tasks out-of-the-box simply given a natural language prompt, which provides the possibility of extracting relations from text without any data and parameter tuning. This work focuses on the study of exploring LLMs, such as ChatGPT, as zero-shot relation extractors. On the one hand, we analyze the drawbacks of existing RE prompts and attempt to incorporate recent prompt techniques such as chain-of-thought (CoT) to improve zero-shot RE. We propose the summarize-and-ask (SUMASK) prompting, a simple prompt recursively using LLMs to transform RE inputs to the effective question answering (QA) format. On the other hand, we conduct comprehensive experiments on various benchmarks and settings to investigate the capabilities of LLMs on zero-shot RE. Specifically, we have the following findings: (i) SUMASK consistently and significantly improves LLMs performance on different model sizes, benchmarks and settings; (ii) Zero-shot prompting with ChatGPT achieves competitive or superior results compared with zero-shot and fully supervised methods; (iii) LLMs deliver promising performance in extracting overlapping relations; (iv) The performance varies greatly regarding different relations. Different from small language models, LLMs are effective in handling challenge none-of-the-above (NoTA) relation.

## 1 Introduction

Relation extraction (RE) aims to identify the relationships between entities in texts, and plays an important role in information extraction (IE). Most of existing RE methods (Zeng et al., 2014; dos Santos et al., 2015) require large amounts of labeled training data which is labor-intensive and time-consuming in practice. Hence, extracting relations from texts using zero or few-shot methodologies has garnered significant scholarly attention (Han et al., 2018; Chen and Li, 2021).

Recent studies (Wei et al., 2022; Wang et al., 2023b) on large-scale pre-trained language models (LLMs), such as GPT-3 (Brown et al., 2020), demonstrate that LLMs perform well in various downstream tasks without any training or fine-tuning but only with a few examples as instructions, which is called *in-context learning*. However, there is currently no consensus on whether LLMs are good few-shot information extractors (Agrawal et al., 2022; Jimenez Gutierrez et al., 2022). Different with some other tasks, RE is more challenging for LLMs because the structured data containing multiple dependent elements are difficult to extract directly and accurately. Although recent studies (Wang et al., 2023a) indicate that some conventional fine-tuning models still outperform LLMs in few-shot RE tasks, we still want to explore whether LLMs can achieve competitive performance compared to fine-tuning models.

Similar to few-shot learning, LLMs also show promising performance on zero-shot settings (Kojima et al., 2022). Recent work for zero-shot RE via prompting LLMs has achieved remarkable progress. QA4RE (Zhang et al., 2023) is a multiple-choice question answering prompt format, in which each relation is transformed into a template and LLMs are expected to predict only a single letter. This prompt is simple but requires manually crafted templates and unable to deal with overlapping relations, which motivates us to find more general and effective prompts. ChatIE (Wei et al., 2023) transforms the zero-shot IE task into a multi-turn question answering problem with a two-stage framework, even surpasses some full shot models on several datasets. Nonetheless, ChatIE still performs worse than the state-of-the-art and is only evaluated on limited benchmarks. Thus it is still unclear how to improve extracting performance by

designing effective prompts and whether LLMs are good zero-shot relation extractors. To this end, we revisit and investigate the potential of LLMs in zero-shot RE on the following research questions:

- **(RQ1)** How does LLMs perform on RE incorporating existing prompt techniques?

- **(RQ2)** How does LLMs perform on zero-shot relation classification?

- **(RQ3)** How does LLMs perform on zero-shot overlapping relation extraction?

Previous work (Ma et al., 2023; Wang et al., 2023a) fails to achieve promising results on RE since black box LLMs such as ChatGPT are difficult to ensure the reliability of outputs. To answer the first question, we investigate the feasibility of incorporating recent prompt techniques to improve the reliability of extracted results. For example, chain-of-thought (CoT) prompting (Wei et al., 2022) improves the reliability of model output by providing intermediate reasoning steps. Active prompting (Diao et al., 2023) is an uncertainty-based active learning method to quantify the uncertainty so as to select the most uncertain outputs. Specifically, we propose the summarize-and-ask (SUMASK) prompting, which decomposed RE into two subtasks: text summarization (Liu and Lapata, 2019) and question answering (Chen et al., 2017a). We further introduce an uncertainty estimation method to approximately characterize output probabilities of LLMs, which yields substantial improvements compared to VANILLA prompting.

To answer the last two questions, we evaluate LLMs on both zero-shot relation classification and overlapping relation extraction. And six RE benchmarks are used for evaluation: (1) FewRel (Han et al., 2018) and Wiki-ZSL (Chen and Li, 2021) for comparision with zero-shot RE methods; (2) TACRED (Zhang et al., 2017), TACREV (Alt et al., 2020) and Re-TACRED (Stoica et al., 2021) for comparision with fully supervised methods; (3) NYT (Riedel et al., 2010) for evaluating overlapping relation extraction. Experimental results demonstrate that LLMs achieve promising results in all experimental settings. In summary, the contributions of this work are three-fold:

- We propose the SUMASK prompting and evaluate the effectiveness of this method. SUMASK consistently and significantly improves LLMs performance by 5.2% - 48.3%

in F1-score compared to VANILLA prompting w.r.t. diverse experimental settings.

- We comprehensively evaluate the capabilities of LLMs on zero-shot relation classification. LLMs achieve competitive or superior results compared with state-of-the-art zero-shot and fully supervised methods. Notably, ChatGPT with SUMASK prompting outperforms the state-of-the-art fully supervised method on TACRED by an average of 2.8% micro-F1.

- We investigate the capabilities of LLMs in extracting overlapping relations. LLMs deliver consistently promising performance encountering different number of triples and various overlapping patterns.

## 2 Related Work

**Few and Zero-shot Relation Extraction** Few-shot RE (Han et al., 2018) aims to predict novel relations by exploring a few labeled instances. Prototypical networks (Snell et al., 2017) are widely used and combined with pre-trained language models (Devlin et al., 2019) in few-shot settings to achieve impressive results. To be capable of extracting relations that were not specified in advance, zero-shot RE (Levy et al., 2017) is proposed to invent new models to predict new relations. However, existing zero-shot methods (Chen and Li, 2021) still requires much labeled data. Recent studies (Zhang et al., 2023; Wei et al., 2023) leverage the LLMs with zero-shot prompting to extract relations from texts without any labeled samples in advance. But it is still unclear whether LLMs are good zero-shot relation extractors by carefully designed prompts. Thus this work aims to investigate the capabilities of LLMs in zero-shot RE.

**Large Language Models and Prompting** Besides the "pre-train and fine-tune" paradigm (Liu et al., 2023), pre-trained LLMs possess characteristics that are advantageous for few-shot (Brown et al., 2020) and zero-shot (Kojima et al., 2022) learning, whereby appropriate prompts are used to effectively guide the model towards generating desired task outputs, thus beginning an era of "pre-train and prompt" (Liu et al., 2021). Prior works (Zhao et al., 2021; Liu et al., 2021) note the sensitivity of prompting under slight modifications. Empirical results demonstrate that answering the restrictive prompts is challenging due to biases acquired during pre-training (Zhao et al., 2021; Arora

et al., 2023). In this study, we evaluate different prompt formats tailored to the particularities of RE and propose SUMASK prompting which outperforms VANILLA prompting by a large margin.

## 3 Problem Definition

Previous zero-shot RE (Chen and Li, 2021) only involves single relation classification. We extend this zero-shot setting to multiple entities and relations. Given the pre-defined relation set $\mathcal{R} = \{r_1, r_2, ..., r_N\}$ and the sentence $\mathcal{S}$ containing the entity set $\mathcal{E} = \{e_1, e_2, ..., e_M\}$, we aim to extract all the relations between these entities composing the relational triples set $\mathcal{Z} = \{(e_i, r_k, e_j)\}$, where $N$ denotes the number of relations, $M$ represents the number of entities, and $e_i, e_j \in \mathcal{E}, r_k \in \mathcal{R}$.

## 4 Prompt Design

### 4.1 VANILLA Prompting

Previous work (Ma et al., 2023; Wang et al., 2023a) claims that LLMs achieve poor results on IE tasks such as RE. We argue that one important reason why LLMs underperform the state-of-the-art is the poor prompt design, as different prompts towards same tasks can cause large variations in the model predictions (Zhao et al., 2021; Arora et al., 2023). Figure 1 illustrates the most direct and common prompt strategy which directly asks LLMs to extract relation labels from text through instructions. However, we empirically find that this approach is ineffective because it makes LLMs to accomplish three no-trivial reasoning processes in only one step: (i) Extracting the relation semantics between the subject and object in the sentence; (ii) Understanding the semantics of each relation label; (iii) Matching the relation semantics between the entities and the given relation labels. Consistent with existing findings (Jimenez Gutierrez et al., 2022; Ma et al., 2023; Wang et al., 2023a), LLMs using VANILLA prompting are unable to achieve satisfactory performance on zero-shot RE.

### 4.2 SUMASK Prompting

Due to the difficulty of LLMs in completing three reasoning processes in one step, we leverage the idea of CoT (Wei et al., 2022) and suggest decomposing this step to artificially guide LLMs in understanding and reasoning. To classify the relations between the subject and object, a simple method is to sequentially ask LLMs whether each relation exists between two entities. For the three reasoning

---

Given the possible relations: [*member of, field of work, work location, ..., father, sibling*].
What are the relations between the subject entity and the object entity expressed by the sentence?
**Sentence:** Savi was born in Pisa, son of Gaetano Savi, professor of Botany at the University of Pisa.
**Subject:** Gaetano Savi
**Object:** Botany
**Relation:** field of work

Figure 1: Illustration of the VANILLA prompting. The output of LLMs is highlighted in color.

processes mentioned above, we design the prompt illustrated in Figure 2 as three steps: (i) Summarize the relations between subject and object given the [INPUT] so as to extract the relation semantics between the two and obtain the intermediate result [SUMMARIZATION]; (ii) Ask LLMs to generate the yes/no questions based on possible triples so as to transform the abstract relation labels to the natural relation descriptions and obtain the intermediate result [QUESTION]; (iii) Ask LLMs to answer the [QUESTION] based on the [SUMMARIZATION] to match the relation semantics between the entities and relations. Then we get the final [ANSWER].

**Uncertainty Estimation** Generally, relation classification assumes that only one correct relation is extracted. However, SUMASK prompt possibly obtains multiple "yes" while querying all the relations in $\mathcal{R}$. Therefore, the final predicted relation is required to select from multiple candidates. Given the sentence $\mathcal{S}$, the subject $e_s$ and object $e_o$, the predicted relation is obtained by:

$$r = \arg\max_{r_i, r_i \in \mathcal{R}} p\left(r_i \mid \mathcal{S}, e_s, e_o\right) \tag{1}$$

Then we aim to transform the multi classification form into multiple binary classification forms. Hence, we define a random variable $\bar{r}$ as follows:

$$\bar{r} = \begin{cases} (1, 0, ..., 0) & r = r_1 \\ (0, 1, ..., 0) & r = r_2 \\ ... \\ (0, 0, ..., 1) & r = r_N \end{cases} \tag{2}$$

Here we make an assumption that only one positive label exists among $N$ binary classification. We denote the intermediate results summarization and question as $s_i$ and $q_i$ corresponding to relation $r_i$.

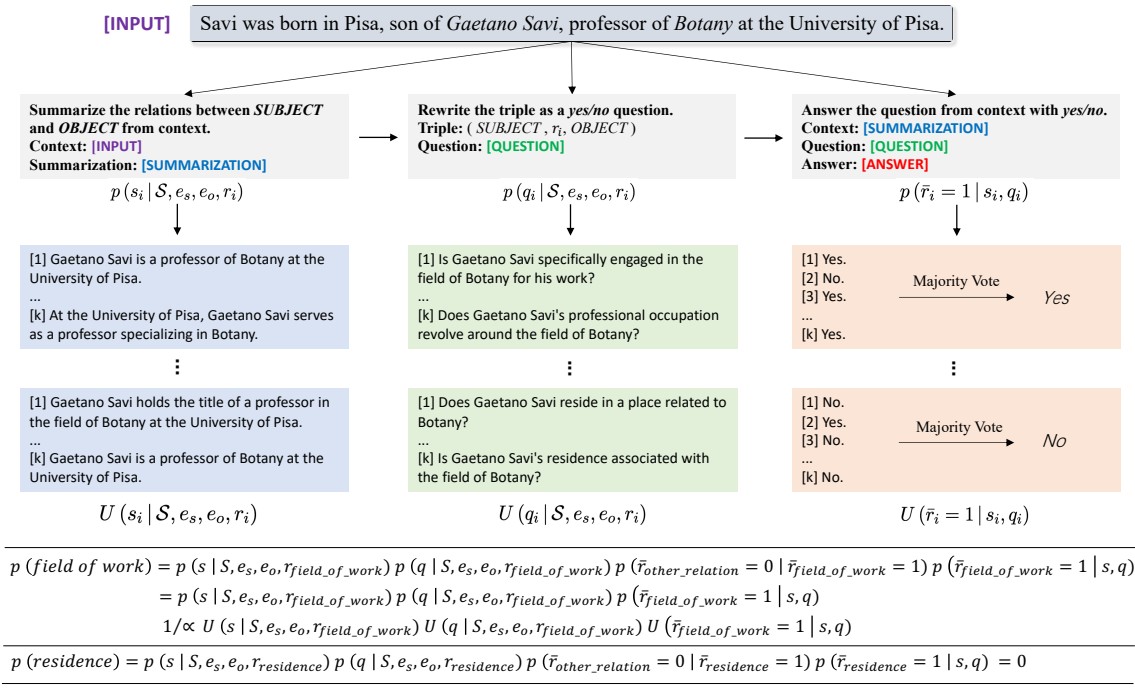

[INPUT] Savi was born in Pisa, son of *Gaetano Savi*, professor of *Botany* at the University of Pisa.

**Summarize the relations between *SUBJECT* and *OBJECT* from context.**
Context: [INPUT]
Summarization: [SUMMARIZATION]

$p(s_i \mid \mathcal{S}, e_s, e_o, r_i)$

**Rewrite the triple as a *yes/no* question.**
Triple: ( *SUBJECT*, $r_i$, *OBJECT* )
Question: [QUESTION]

$p(q_i \mid \mathcal{S}, e_s, e_o, r_i)$

**Answer the question from context with *yes/no*.**
Context: [SUMMARIZATION]
Question: [QUESTION]
Answer: [ANSWER]

$p(\bar{r}_i = 1 \mid s_i, q_i)$

[1] Gaetano Savi is a professor of Botany at the University of Pisa.
...
[k] At the University of Pisa, Gaetano Savi serves as a professor specializing in Botany.

[1] Is Gaetano Savi specifically engaged in the field of Botany for his work?
...
[k] Does Gaetano Savi's professional occupation revolve around the field of Botany?

[1] Yes.
[2] No.
[3] Yes.
...
[k] Yes.
Majority Vote → Yes

[1] Gaetano Savi holds the title of a professor in the field of Botany at the University of Pisa.
...
[k] Gaetano Savi is a professor of Botany at the University of Pisa.

[1] Does Gaetano Savi reside in a place related to Botany?
...
[k] Is Gaetano Savi's residence associated with the field of Botany?

[1] No.
[2] Yes.
[3] No.
...
[k] No.
Majority Vote → No

$U(s_i \mid \mathcal{S}, e_s, e_o, r_i)$

$U(q_i \mid \mathcal{S}, e_s, e_o, r_i)$

$U(\bar{r}_i = 1 \mid s_i, q_i)$

$$p(\text{field of work}) = p(s \mid S, e_s, e_o, r_{field\_of\_work}) \, p(q \mid S, e_s, e_o, r_{field\_of\_work}) \, p(\bar{r}_{other\_relation} = 0 \mid \bar{r}_{field\_of\_work} = 1) \, p(\bar{r}_{field\_of\_work} = 1 \mid s, q)$$
$$= p(s \mid S, e_s, e_o, r_{field\_of\_work}) \, p(q \mid S, e_s, e_o, r_{field\_of\_work}) \, p(\bar{r}_{field\_of\_work} = 1 \mid s, q)$$
$$1/\propto U(s \mid S, e_s, e_o, r_{field\_of\_work}) \, U(q \mid S, e_s, e_o, r_{field\_of\_work}) \, U(\bar{r}_{field\_of\_work} = 1 \mid s, q)$$

$$p(\text{residence}) = p(s \mid S, e_s, e_o, r_{residence}) \, p(q \mid S, e_s, e_o, r_{residence}) \, p(\bar{r}_{other\_relation} = 0 \mid \bar{r}_{residence} = 1) \, p(\bar{r}_{residence} = 1 \mid s, q) = 0$$

Figure 2: Illustration of the SUMASK prompting. The outputs of LLMs are highlighted in color. The probability of relation "residence" is 0 because the system answers "no" via majority vote. To estimate the uncertainty of relation "field of work", we generate $k$ [SUMMARIZATION], [QUESTION], [ANSWER] representations, respectively. Then we calculate the dispersion degree among these representations to approximate the uncertainty.

Then the probability of relation $r_i$ is obtained by:

$$
\begin{aligned}
p(r_i) &= p(\bar{r} \mid \mathcal{S}, e_s, e_o, r_i) \\
&= p(\bar{r} \mid s_i, q_i) \, p(s_i, q_i \mid \mathcal{S}, e_s, e_o, r_i) \\
&= p(\bar{r}_{-i} = 0 \mid \bar{r}_i = 1) \, p(\bar{r}_i = 1 \mid s_i, q_i) \\
&\quad p(q_i \mid \mathcal{S}, e_s, e_o, r_i) \, p(s_i \mid \mathcal{S}, e_s, e_o, r_i)
\end{aligned}
\tag{3}
$$

where $r_i = 1$ indicates that LLMs answer "yes" based on the summarization and question of relation $i$. Based on the above one positive label assumption, we have:

$$
p(\bar{r}_{-i} = 0 \mid \bar{r}_i = 1) = \begin{cases} 1 & \bar{r}_i = 1 \\ 0 & \bar{r}_i = 0 \end{cases}
\tag{4}
$$

The final predicted relation is selected from all the candidate relations with the max probability:

$$
r = \operatorname*{arg\,max}_{r_i, r_i \in \mathcal{R}} p(\bar{r}_i = 1 \mid s_i, q_i) \\
p(q_i \mid \mathcal{S}, e_s, e_o, r_i) \, p(s_i \mid \mathcal{S}, e_s, e_o, r_i)
\tag{5}
$$

Unfortunately, it is difficult to get the conditional probability of each step in LLMs. For instance, the "gpt-3.5-turbo" model only provides the final natural text output without any logit or probability. To this end, we introduce an uncertainty estimation method to approximately characterize conditional

probabilities. Finding the relation $r$ that satisfies equation 5 is equivalent to:

$$
\begin{aligned}
r = \operatorname*{arg\,min}_{r_i, r_i \in \mathcal{R}} \; & U(\bar{r}_i = 1 \mid s_i, q_i) \\
& U(q_i \mid \mathcal{S}, e_s, e_o, r_i) \, U(s_i \mid \mathcal{S}, e_s, e_o, r_i)
\end{aligned}
\tag{6}
$$

where $U(X|Y)$ represents the uncertainty of the random variable $X$ under the known random variable $Y$. Therefore, the relation with the smallest uncertainty is selected as final prediction.

Inspired by Diao et al. (2023), we consider measuring the uncertainty using the dispersion degree among $k$ generated answers $A = \{a_1, ..., a_k\}$ as shown in Figure 2. Specifically, we feed answers $A$ into a pre-trained Sentence-BERT encoder (Reimers and Gurevych, 2019) to generate the answer representations $\mathbf{Z} = \{\mathbf{z}_1, ..., \mathbf{z}_k\}$. Then the uncertainty is calculated by:

$$
u = \frac{1}{k-1} \sum_{i=1}^{k} d\left(\mathbf{z}_i, \frac{1}{k} \sum_{j=1}^{k} \mathbf{z}_j\right)
\tag{7}
$$

where $d(\cdot)$ function measures the distance between two representations. After obtaining the uncer-

tainty of each step, we select the relation $r$ via:

$$r = \underset{r_i, r_i \in \mathcal{R}}{\arg\min} \, u_1 \cdot u_2 \cdot u_3 \tag{8}$$

$$u_1 \propto U\left(s_i \,|\, \mathcal{S}, e_s, e_o, r_i\right) \tag{9}$$

$$u_2 \propto U\left(q_i \,|\, \mathcal{S}, e_s, e_o, r_i\right) \tag{10}$$

$$u_3 \propto U\left(\bar{r}_i = 1 \,|\, s_i, q_i\right) \tag{11}$$

We adopt the majority vote (Wang et al., 2023b) to determine the yes/no answer in last step. If the system answers "no" with every relation $r_i \in \mathcal{R}$, the prediction is NoTA. For overlapping relations, we simply consider all relations that answer with "yes" as predictions.

**Entity-Relation Mapping**  Obviously, asking LLMs for each relation is inefficient. Inspired by Li et al. (2022), we adopt the entity-relation mapping mechanism to deal with relation redundancy. Specifically, when the entity type is determined, the relations that possibly related to it are also determined, so most impossible relations are discarded in advance. Note that the VANILLA prompting also adopts this simple strategy. This simple mechanism not only improves efficiency but also benefits overall performance.

## 5   Experiments

### 5.1   Datasets

**Simple Relation Classification**  We evaluate LLMs in zero-shot simple relation classification on FewRel (Han et al., 2018), Wiki-ZSL (Chen and Li, 2021), TACRED (Zhang et al., 2017), TACREV (Alt et al., 2020) and Re-TACRED (Stoica et al., 2021). For FewRel and Wiki-ZSL, we follow the previous work (Chen and Li, 2021) and randomly select $m$ = 5, 10, 15 unseen relations. For TACRED, TACREV and Re-TACRED, we evaluate the LLMs on its test set. Note that there is no entity type information provided in FewRel and Wiki-ZSL while entity types become available in TACRED, TACREV and Re-TACRED. We adopt the precision, recall and macro-F1 for FewRel and Wiki-ZSL (Chen and Li, 2021), while micro-F1 is used for TACRED, TACREV and Re-TACRED (Zhou and Chen, 2021).

**Overlapping Relation Extraction**  We adopt the NYT (Riedel et al., 2010) to test the ability of extracting overlapping relations. For NYT, we assume the entities and their types in the sentence are available and models only extract the overlapping

relations given the entities in its test set. We use micro-F1 for evaluation. NYT is only evaluated on LLMs as existing baselines have not considered this multiple entities and relations zero-shot setting.

To keep OpenAI API costs under control, we randomly select 1,000 samples in the corresponding test set according to the proportion of samples in each relation class. Specifically, the number of samples corresponding to each relation in FewRel is the same. Note that 78.56% of samples in TACRED test set belongs to NoTA relation, while it is 57.91% in Re-TACRED. We provide the statistics of the datasets in Appendix A.

### 5.2   Baselines

**Zero-shot Baselines**  For FewRel and Wiki-ZSL, we choose R-BERT (Wu and He, 2019), ESIM (Chen et al., 2017b), CIM (Rocktäschel et al., 2016) and ZS-BERT (Chen and Li, 2021) as the zero-shot RE baselines. Note that Relation-Prompt (Chia et al., 2022) uses the seq2seq-based models to generate pseudo data of unseen relations to fine-tune the model. And recent method RE-Matching (Zhao et al., 2023) requires elaborated relation descriptions to achieve superior performance but the open source code is not yet available. Thus the two methods are not discussed in this study.

**Supervised Baselines**  For TACRED, TACREV and Re-TACRED, fully supervised models such as PA-LSTM (Zhang et al., 2017), C-GCN (Zhang et al., 2018), SpanBERT (Joshi et al., 2020), LUKE (Yamada et al., 2020), NLI-DeBERTa (Sainz et al., 2021), SuRE-PEGASUS (Lu et al., 2022) and DeepStruct (Wang et al., 2022) are selected to compare with our zero-shot prompt based methods. We also test NLI-DeBERTa, SuRE-PEGASUS, DeepStruct and QA4RE (Zhang et al., 2023) under zero-shot setting to investigate the NoTA relation impact.

**LLMs Baselines**  We investigate open source LLMs such as GPT-J (Wang and Komatsuzaki, 2021), BLOOM (Scao et al., 2022) and T0 (Sanh et al., 2022) with SUMASK prompting with other state-of-the-art models in zero-shot settings. For the parameter scale, we choose GPT-J-6B [1], BLOOM-7.1B [2] and T0pp-11B [3] for experiments. For ChatGPT (Ouyang et al., 2022), we use the

---

[1]https://huggingface.co/EleutherAI/gpt-j-6b
[2]https://huggingface.co/bigscience/bloom
[3]https://huggingface.co/bigscience/T0

| Datasets | FewRel m=5 | | | Wiki-ZSL m=5 | | |
|---|---|---|---|---|---|---|
| | P | R | F1 | P | R | F1 |
| R-BERT | 42.19 | 48.61 | 45.17 | 39.22 | 43.27 | 41.15 |
| ESIM | 56.27 | 58.44 | 57.33 | 48.58 | 47.74 | 48.16 |
| CIM | 58.05 | 61.92 | 59.92 | 49.63 | 48.81 | 49.22 |
| ZS-BERT | 76.96 | **78.86** | **77.90** | 71.54 | **72.39** | 71.96 |
| GPT-J | 40.75 | 45.63 | 43.05 | 40.23 | 46.94 | 43.33 |
| BLOOM | 43.62 | 48.15 | 45.77 | 41.97 | 45.38 | 43.61 |
| T0 | 43.05 | 54.97 | 48.29 | 42.16 | 53.92 | 47.32 |
| VANILLA | 67.41 | 72.97 | 70.08 | 64.47 | 70.83 | 67.50 |
| SUMASK | **78.27** | 72.55 | 75.30 | **75.64** | 70.96 | **73.23** |

| Datasets | m=10 | | | m=10 | | |
|---|---|---|---|---|---|---|
| | P | R | F1 | P | R | F1 |
| R-BERT | 25.52 | 33.02 | 28.20 | 26.18 | 29.69 | 27.82 |
| ESIM | 42.89 | 44.17 | 43.52 | 44.12 | 45.46 | 44.78 |
| CIM | 47.39 | 49.11 | 48.23 | 46.54 | 47.90 | 45.57 |
| ZS-BERT | 56.92 | 57.59 | 57.25 | 60.51 | 60.98 | 60.74 |
| GPT-J | 28.37 | 32.27 | 30.19 | 27.13 | 32.76 | 29.68 |
| BLOOM | 29.28 | 33.81 | 31.38 | 29.45 | 34.19 | 31.64 |
| T0 | 29.87 | 34.26 | 31.91 | 30.18 | 35.48 | 32.62 |
| VANILLA | 42.48 | 46.26 | 44.29 | 41.83 | 46.22 | 43.92 |
| SUMASK | **64.77** | **60.94** | **62.80** | **62.31** | **61.08** | **61.69** |

| Datasets | m=15 | | | m=15 | | |
|---|---|---|---|---|---|---|
| | P | R | F1 | P | R | F1 |
| R-BERT | 16.95 | 19.37 | 18.08 | 17.31 | 18.82 | 18.03 |
| ESIM | 29.15 | 31.59 | 30.32 | 27.31 | 29.62 | 28.42 |
| CIM | 31.83 | 33.06 | 32.43 | 29.17 | 30.58 | 29.86 |
| ZS-BERT | 35.54 | 38.19 | 36.82 | 34.12 | 34.38 | 34.25 |
| GPT-J | 20.36 | 35.00 | 25.74 | 20.83 | 34.37 | 25.94 |
| BLOOM | 22.62 | 36.45 | 27.92 | 22.37 | 34.26 | 27.07 |
| T0 | 24.05 | 36.83 | 29.09 | 23.16 | 34.90 | 27.84 |
| VANILLA | 25.71 | 27.77 | 26.70 | 23.17 | 27.82 | 25.28 |
| SUMASK | **44.76** | **41.13** | **42.87** | **43.55** | **40.27** | **41.85** |

Table 1: Main results on FewRel and Wiki-ZSL. In order to reduce the effect of experimental noise, the unseen label selection process is repeated for five different random seeds to produce the test set. The results of the baselines are retrieved from Chen and Li (2021).

| Datasets | TACRED | TACREV | Re-TACRED |
|---|---|---|---|
| PA-LSTM | 65.1 | 73.3[‡] | 79.4[†] |
| C-GCN | 66.3 | 74.6[‡] | 80.3[†] |
| SpanBERT | 70.8 | 78.0[*] | 85.3[†] |
| LUKE | 72.7 | 80.6[‡] | **90.3**[‡] |
| NLI-DeBERTa | 73.9 | - | - |
| SuRE-PEGASUS | 75.1 | **83.3** | - |
| DeepStruct | 76.8 | - | - |
| GPT-J | 44.4 | 40.7 | 38.3 |
| BLOOM | 46.5 | 41.2 | 40.8 |
| T0 | 59.0 | 57.5 | 55.5 |
| VANILLA | 31.3 | 30.4 | 28.0 |
| SUMASK | **79.6** | 75.1 | 73.8 |

Table 2: Micro-F1 score on TACRED, TACREV and Re-TACRED. ∗ marks re-implemented results from Alt et al. (2020). † marks re-implemented results from Stoica et al. (2021). ‡ marks re-implemented results from Zhou and Chen (2021). Others are retrieved from original papers.

| Methods | Micro-P | Micro-R | Micro-F1 | Macro-F1 |
|---|---|---|---|---|
| SuRE-PEGASUS | 13.8 | 51.7 | 21.8 | 14.9 |
| NLI-DeBERTa | 42.9 | 76.9 | 55.1 | 55.0 |
| DeepStruct† | 32.7 | 40.6 | 36.2 | 32.8 |
| QA4RE | 47.7 | **78.6** | **59.4** | **58.9** |
| VANILLA | 33.8 | 39.2 | 36.3 | 27.4 |
| SUMASK | **62.2** | 53.8 | 57.7 | 57.9 |

Table 3: NoTA-excluded 41-class micro-F1 and NoTA-included 42-class macro F1 on TACRED. † marks our re-implementation results. The rest results of the baselines are retrieved from Zhang et al. (2023).

"gpt-3.5-turbo-0301", which is the most capable GPT-3.5 model and optimized for chat. We denote the combination of ChatGPT and two prompts as VANILLA and SUMASK for brevity. Similar to Kojima et al. (2022), after the model outputs a text, our method picks up only the part of the answer text that first satisfies the answer format. The implementation details are provided in Appendix B.

## 5.3 Relation Classification Results

**Main Results** The results by varying $m$ unseen relations on FewRel and Wiki-ZSL are summarized in Table 1. Generally, LLMs with zero-shot prompting achieve competitive results compared to existing zero-shot RE methods over two datasests when targeting at different numbers of unseen rela-

tions. Specially, the proposed SUMASK prompting makes ChatGPT deliver superior results compared to ZS-BERT in most cases. As $m$ increases, it is straightforward that models are difficult to predict the right relation since the possible choices have increased. The superiority of SUMASK gets more significant when the number of unseen relations increases while VANILLA suffers grave declines. Such results not only validate the effectiveness of proposed prompting, but indicate SUMASK is less sensitive to the number of relations compared to baselines. Moreover, GPT-J-6B, BLOOM-7.1B and T0-11B with SUMASK exceed GPT-3.5-175B with VANILLA, and match the performance of previous text entailment models ESIM and CIM.

The main results on TACRED, TACREV and Re-TACRED are shown in Table 2. Compared to fully supervised methods, zero-shot prompting with LLMs still show competitive results. Notably, ChatGPT with SUMASK prompting outperforms the state-of-the-art fully supervised method Deep-Struct on TACRED by an average of 2.8% micro-

| Best performance | Accuracy | Worst performance | Accuracy |
|---|---|---|---|
| voice type | 97.4 | language of work or name | 13.0 |
| occupation | 97.1 | tributary | 22.7 |
| contains administrative territorial entity | 95.4 | residence | 31.6 |
| participant of | 95.1 | mouth of the watercourse | 33.3 |
| crosses | 95.0 | screenwriter | 42.4 |
| located in the administrative territorial entity | 94.9 | performer | 44.0 |
| league | 94.4 | head of government | 44.3 |
| constellation | 93.9 | father | 45.3 |
| competition class | 93.1 | distributor | 48.4 |
| heritage designation | 93.0 | located on terrain feature | 49.6 |

Table 4: Top-10 relations with best (left) and worst (right) performance.

F1. The interesting finding is that the performance of LLMs decreases on TACREV and Re-TACRED while fine-tuned models steadily improves. The reason might be that the high proportion of NoTA in TACRED makes zero-shot prompting with LLMs surpass fully supervised methods. It is difficult for conventional models to form a good NoTA representation (Han et al., 2020). Jimenez Gutierrez et al. (2022) also demonstrate that the earlier inferior performance of LLMs on RE tasks can be largely attributed to their inability to handle the NoTA relation. To this end, we provide an evaluation of zero-shot methods on NoTA relation. Following previous work (Sainz et al., 2021; Zhang et al., 2023), we report the NoTA-excluded micro-F1 and NoTA-included macro-F1 to investigate the extracting ability of normal and NoTA relation.

The NoTA relation results are shown in Table 3. First, SUMASK prompting is not prominent in 41 semantic relations, which demonstrate the high micro-F1 score is mainly due to the high portion of NoTA relation. Second, SUMASK also achieves significant improvement like QA4RE in NoTA-included metrics compared to the small LM-based NLI methods. For VANILLA, excluding NoTA relation brings better results. This further demonstrates the sensitivity of prompting and the effectiveness of proposed prompting.

**Relation Specific Analysis** Due to biases acquired during pre-training, LLMs have different abilities to understand different relations, which leads to varying levels of extraction results. We analyze the performance differences through experiments on 80-relation dataset FewRel. Specifically, under the SUMASK framework, we ask the LLMs whether the answer of question generated by golden triple is "yes". Then we adopt the accuracy metric to evaluate the performance of each relation. Finally, we select 10 relations with the

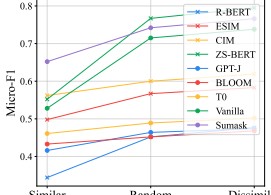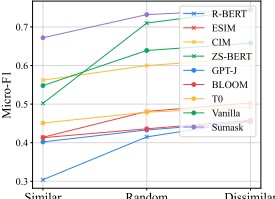

Figure 3: Performance comparison between five similar, random and dissimilar relations.

best and worst performance, as shown in Figure 4. Surprisingly, the accuracy difference between the best ("voice type") and the worst ("language of work or name") relation is 84.4%. We provide the detailed analysis in Appendix C.

The semantic similarity between relations in the embedding space greatly impacts the zero-shot RE performance. Following Chen and Li (2021), we select five semantically distant relations and the other five relations that possess similar semantics to evaluate on our baselines, illustrated in Figure 3. Obviously, dissimilar relations lead to better results. First, when enhanced by SUMASK prompting, LLMs delivers more stable results because of the smaller performance gap between three settings. Second, the text-entailment based methods are less affected by similar relations compared to embedding-based models such as R-BERT and ZS-BERT. Because the predictions by text entailment based methods ESIM, CIM and SUMASK prompting do not resort to similarity search.

**Prompt Strategy Analysis** We study the effectiveness of proposed SUMASK prompting. We conduct the ablation study about summarization generation, question generation and uncertainty estimation. Specifically, we omit the summarization process, replace the LLMs generated questions with pre-defined question templates (Appendix D) and

| Datasets | FewRel | | | TACRED | |
|---|---|---|---|---|---|
| | m=5 | m=10 | m=15 | NoTA | w/o NoTA |
| SUMASK | **75.3** | **62.8** | **42.9** | **79.6** | 51.6 |
| SUMASK w/o Sum. | 67.6 | 58.1 | 40.4 | 76.4 | 48.5 |
| SUMASK w/o Ask. | 71.4 | 56.8 | 37.7 | 78.7 | **54.3** |
| SUMASK w/o Unc. | 38.8 | 29.6 | 23.0 | 76.3 | 25.2 |

Table 5: Results of ablation study.

| Methods | P | R | F1 | N=1 | N=2 | N=3 | N=4 | N>=5 |
|---|---|---|---|---|---|---|---|---|
| GPT-J | 31.4 | 53.6 | 39.6 | 39.5 | 36.9 | 39.3 | 39.8 | 30.3 |
| BLOOM | 35.2 | 56.9 | 43.5 | 39.2 | 38.4 | 42.1 | 44.6 | 33.6 |
| T0 | 39.6 | 57.3 | 46.8 | 44.3 | 41.9 | 47.2 | 48.3 | 35.7 |
| VANILLA | 20.4 | 16.5 | 18.2 | 31.3 | 22.7 | 16.3 | 12.8 | 7.7 |
| SUMASK | **55.7** | **78.3** | **65.1** | **65.6** | **62.5** | **66.7** | **70.8** | **59.5** |

Table 6: Main results on NYT.

Figure 4: The correlations between uncertainty estimation and ground truth.

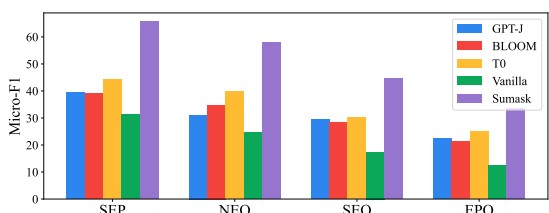

Figure 5: F1-score of extracting overlapping relations from sentences with different overlapping patterns.

randomly select the relation from candidates without uncertainty estimation, respectively. Table 5 shows the ablation study results. Summarization consistently improves the overall performance under different settings, which indicates that incorporating reasoning steps before predicting relation is reasonable. Compared to pre-defined templates, LLMs generated questions may not necessarily be the best choice. Manually designed template enables the semantic description of relations more accurate, but our simple method is convenient and requires no external interference. Uncertainty estimation shows the significant impact on performance. Note that SUMASK still achieves 76.3% F1 on TACRED, because the uncertainty estimation has no impact on NoTA relation theoretically. To understand the rationality of uncertainty estimation, we select 500 samples from each datasets to illustrate the correlations between uncertainty estimation and ground truth. Intuitively, golden relations have relatively low uncertainty. Figure 4 shows that most golden relations correspond to low uncertainty, while only a few correspond to large uncertainty, which is consistent with our intuition.

### 5.4 Overlapping Relation Extraction Results

The overlapping relation extraction results are illustrated in Table 6. VANILLA prompting is difficult to handle the overlapping relations as LLMs always tend to output only one relation. In contrast, SUMASK prompting is transferable and consistent on LLMs with different sizes. Note that different from the relation classification results, the recall of SUMASK is higher than its precision. Because

regarding all the candidate relations as predictions brings many false positives. Setting thresholds for uncertainty estimation might be a feasible solution.

To further study the capability of LLMs in extracting overlapping relations, we conduct experiments on different types of sentences. We split the sentences into five classes and use $N$ to denote the number of relational triples in a sentence. Again, the SUMASK prompting achieves good performance over all five classes. Overlapping relational triples are summarized into four patterns: SEP (*SingleEntityPair*), NEO (*NoEntityOverlap*), SEO (*SingleEntityOverlap*) and EPO (*EntityPairOverlap*). Figure 5 shows that the performance of most baselines on four patterns presents a decreasing trend, reflecting the increasing difficulty of extracting relational triples from sentences with different overlapping patterns, encouraging more consistent and effective methods in the future research.

## 6 Conclusion

This work provides a comprehensive study on zero-shot RE with prompting-based LLMs. Besides the VANILLA prompting, we introduce a novel SUMASK prompting to fully explore the power of LLMs. Our experiments on six benchmarks demonstrate the capability of LLMs in zero-shot RE. Furthermore, we are able to answer the three aforementioned questions. Recent prompt techniques such as CoT significantly improve zero-shot RE prompting. Properly instructed LLMs not only deliver competitive or superior results compared to state-of-the-art relation classification models, but also are promising for zero-shot overlapping RE.

## Limitations

We only carry out comprehensive experiments on zero-shot RE without few-shot and domain-specific exploration. It is still unclear what are the capabilities of LLMs on domain-specific datasets and how much performance could be improved by few-shot prompting. Our limited budget also restricted our study to a small set of prompt styles. It is possible that having a larger prompt design search space could narrow the gap between models fine-tuning and LLMs in-context learning.

## Ethics Statement

In this work, we investigate the capability of LLMs on the important and fundamental task of zero-shot relation extraction. We do not anticipate any ethical issues regarding the topics of this research.

## Acknowledgement

This work was supported by National Science Foundation of China (Grant Nos.62376057), a specialized intelligent science project (2023-110ZTA05) XX Technology for Single and Group Intelligent, a specialized intelligent science project Key entity recognition and analysis technology in XX field.

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

## A Dataset Statistics

The statistics of the datasets are shown in Table 7 and Table 8.

| Dataset | # instances | # entities | # relations |
|---|---|---|---|
| FewRel | 56,000 | 72,954 | 80 |
| Wiki-ZSL | 94,383 | 77,623 | 113 |

Table 7: Statistics of FewRel and Wiki-ZSL.

| Dataset | # train | # dev | # test | # relations |
|---|---|---|---|---|
| TACRED | 68,124 | 22,631 | 15,509 | 42 |
| TACREV | 68,124 | 22,631 | 15,509 | 42 |
| Re-TACRED | 58,465 | 19,584 | 13,418 | 40 |
| NYT | 56,196 | 5,000 | 5,000 | 24 |

Table 8: Statistics of TACRED, TACREV, Re-TACRED and NYT.

## B Implementation Details

For the hyper-parameters of SUMASK prompting, we set the number of generated answers $k$ as 5, and we use the "bert-large-nli-mean-tokens" version of Sentence-BERT encoder to generate the answer representations. For open source LLMs GPT-J, BLOOM and T0, we set the max generated length as 128 and the temperature as 0.3. Note that the results of VANILLA prompting with open source LLMs are not discussed in our paper because they all achieve near-0 performance. Due to the high noise content in the output of LLMs, we pick up only the part of the answer text that first satisfies the answer format to alleviate the unexpected behaviors. For gpt-3.5-turbo-0301, we set the max length as 256 and the temperature as 0.7 according to official default setting. We treat the outputs of ChatGPT as valid results without post-processing.

## C Relation Specific Analysis

We provide accuracy results for all relations in FewRel. The results are summarized in Table 9. And we provide several case studies to analyze the performance of ChatGPT on different relations.

**"language of work or name"** We observe two important reasons for the poor performance of this relation, shown in Table 10. On the one hand, ChatGPT sometimes misunderstand the semantics of entities or relations, which leads to generated questions deviating from the original meaning expressed. For example, Elizabeth is an English female name but ChatGPT treats the name as a person (**Case 2**), which also indicates the drawback of this method that the generated questions might be unexpected without providing the context or template. This also highlights the importance of incorporating entity types into relation extraction. On the other hand, prompting is a brittle process wherein small modifications to the prompt can cause large variations in the model predictions. For example, we use "Answer the question from context" rather "Answer the question from context with *yes/no*" to expect that ChatGPT could not only give the "yes/no" answer but also provide the reason for its judgment. We achieve the expected results in most relations. However, the answers corresponding to the relation "language of work or name" frequently do not contain "yes/no" while express positive (**Case 3**), which makes automatic evaluation difficult. Therefore, the specific form of the statement answer is important, otherwise it may lead to unreasonable evaluation. Moreover, the annotation errors (**Case 4**) of this relation also lead to biased and unreliable evaluation.

**"tributary" and "mouth of the watercourse"** ChatGPT performs poorly in both two relations. First, these two relations have very similar semantics because they are reciprocal in FewRel, as shown in **Case 1**. Unfortunately, ChatGPT frequently reverses the subject and object corresponding to these two relations during question generation step. Specifically, ChatGPT treats the triple (lisava river, tributary, natra river) as (lisava river, tributary of, natra river). This phenomenon highlights the advantage of manually crafted question templates and persuades us to provide few-shot demonstrations to generate reliable questions. Second, we also find plenty of annotation errors of two relations because of distant supervision, shown in **Case 2**. We can see that ChatGPT provides the correct judgments and reasons for this case, which makes it possible for LLMs to become reliable annotation inspectors.

## D Question Templates

In ablation study, we replace the LLMs generated questions with pre-defined question templates. For FewRel, we simply define a template that forms with "The relation between '*subject*' and '*object*' is '*relation name*'. Yes or No?". For TACRED, we follow Zhang et al. (2023) to use the templates shown in Table 12.

## E  Discussions

**The empirical validation of SUMASK prompting** The underlying operational mechanisms intrinsic to SUMASK prompting rely on the strong capabilities of LLMs as zero-shot reasoners. Following the prompt instructions, LLMs pay attention to the entities of interest, infer and summarize the relations between them. To the best of our knowledge, the logical reasoning faculties of LLMs are not explicitly utilized in previous work of zero-shot relation extraction. From raw text inputs to extracted relation labels, this process lacks intermediate reasoning steps. We decompose the zero-shot relation extraction into three steps to make LLMs sensitize to the semantic understanding and logical reasoning. With the proper instruction for summarization, LLMs are able to perform logical reasoning on specific entities and obtain relations between them. The LLMs can automatically do this via prompting, but the small fine-tuned model cannot. **SUMASK elicits the logical reasoning ability inside LLMs for relation deduction**. The ablation results also show that without the summarization step, the overall performance drops 2.5% - 7.7% F1 on FewRel and TACRED under different settings. The experimental results on overlapping relation extraction also demonstrate the superior of SUMASK prompting over VANILLA prompting.

Here we provide a case study on NYT. The input prompt is: ***Summarize the relations between "Cambodia" and "Penh" from context. Context: Homage to Cambodia was performed at Chaktomuk Conference Hall in Phnom Penh on Oct. 21 , attended by the king. Summarization:***

Then the response from ChatGPT is ***"Phnom Penh" is the capital city of "Cambodia," where the event "Homage to Cambodia" took place at the "Chaktomuk Conference Hall" on October 21.*** Obviously, the first sentence *"Phnom Penh" is the capital city of "Cambodia,"* generated by LLMs clearly elucidates the relationship between two entities, facilitating the subsequent processes.

**The advantages of SUMASK prompting** Generally, the SUMASK prompting does not require any sort of prompt engineering or template writing to start using. And this is one of our contribution points. Specially, not all relations can be accurately described by templates because the description of relations may vary across different entities. For instance, the relation ***language of work or name***

in datast FewRel is hard to describe by a single template, while this relation can not only describe language versions of some literary works, but also describe what language a name belongs to. Consider the following two templates: (1) **The language of *subject* is *object*.** and (2) ***subject* is a name in *object*.** The triple *(Elizabeth, language of work or name, English)* satisfy the second template but deliver confusions in the first template, while the triple *(The Lord of the Rings, language of work or name, English)* only satisfy the first template. **SUMASK does not require any sort of prompt engineering or template writing to start using**.

**The complexity of SUMASK prompting** Typically, the SUMASK prompting method suffers from relatively high inference complexity as it needs to enumerate all possible triples to obtain summarizations, questions, and answers, for $k$ times. Suppose we have $n$ samples to be extracted and $r$ candidate relations, the total complexity is $O(k \times r \times n)$. First, using entity types to discard the most irrelevant relations is a useful method, which achieves less complexity $O(k \times \hat{r} \times n)$ where $\hat{r}$ represents the maximum value of the mapped relation candidate set and much less than $r$. Second, we can certainly ask multiple samples (not exceeding the model maximum length) to LLMs at one time to improve inference speed. Suppose we concatenate $k$ samples in a prompt, then the complexity becomes $O(\hat{r} \times n)$. More efficient zero-shot prompting for relation extraction is worth exploring in the future.

**Uncertainty estimation in all LLMs** Uncertainty estimation is based on an assumption that the outputs of LLMs would be stabler with the predictions equivalent to the ground truth. The efficiency of using logits to generate the probabilities of intermediate results is relatively low, because we are required to obtain the probability of tokens at each position. Moreover, it is highly susceptible to extreme values. For example, if a token with a low probability is sampled during sampling process, it will affect the probability value of the entire sentence. In addition, due to the generated long text sequence, the difference of probability values between generated sentences is not obvious, making it difficult to choose the relation with the highest probability. Using SUMASK with uncertainty estimation brings better outcomes and we show this technique is suitable for both white box (e.g., BLOOM) and black box (e.g., ChatGPT) models.

| Relation Name | Accuracy | Relation Name | Accuracy |
|---|---|---|---|
| voice type | 97.4 | after a work by | 77.4 |
| occupation | 97.1 | mountain range | 76.6 |
| contains administrative territorial entity | 95.4 | composer | 76.3 |
| participant of | 95.1 | operating system | 76.1 |
| crosses | 95.0 | notable work | 76.0 |
| located in the administrative territorial entity | 94.9 | sibling | 75.6 |
| league | 94.4 | developer | 73.3 |
| constellation | 93.9 | located in or next to body of water | 71.1 |
| competition class | 93.1 | record label | 69.7 |
| heritage designation | 93.0 | follows | 69.0 |
| position held | 92.7 | original language of film or TV show | 68.7 |
| member of political party | 92.7 | operator | 67.9 |
| location | 92.0 | location of formation | 67.4 |
| field of work | 92.0 | country of origin | 66.4 |
| part of | 91.6 | successful candidate | 66.1 |
| has part | 91.0 | country of citizenship | 65.9 |
| military branch | 91.0 | subsidiary | 65.9 |
| instance of | 90.1 | licensed to broadcast to | 64.1 |
| sport | 88.6 | main subject | 63.6 |
| applies to jurisdiction | 88.1 | owned by | 62.6 |
| member of | 88.1 | mother | 62.3 |
| participating team | 88.1 | occupant | 61.9 |
| taxon rank | 87.9 | position played on team / speciality | 60.9 |
| characters | 87.6 | followed by | 59.4 |
| genre | 86.1 | said to be the same as | 59.3 |
| instrument | 86.0 | place served by transport hub | 59.1 |
| director | 85.9 | sports season of league or competition | 58.7 |
| participant | 85.6 | child | 57.9 |
| manufacturer | 85.1 | headquarters location | 56.6 |
| country | 84.3 | work location | 53.0 |
| movement | 84.3 | located on terrain feature | 49.6 |
| architect | 84.0 | distributor | 48.4 |
| winner | 83.9 | father | 45.3 |
| original broadcaster | 82.9 | head of government | 44.3 |
| religion | 82.1 | performer | 44.0 |
| nominated for | 81.4 | screenwriter | 42.4 |
| platform | 80.3 | mouth of the watercourse | 33.3 |
| military rank | 79.9 | residence | 31.6 |
| publisher | 79.7 | tributary | 22.7 |
| spouse | 79.6 | language of work or name | 13.0 |

Table 9: Accuracy of each relation in FewRel.

**Case 1**
Summarize the relations between "brut y brenhinedd" and "welsh" from context.
Context: Walter 's name is attached to the " brut tysilio " , a variant of the welsh chronicle " brut y brenhinedd ".
Summarization: Walter is associated with the brut tysilio, which is a welsh chronicle known as brut y brenhinedd.

Rewrite the triple as a yes/no question.
Triple: (brut y brenhinedd, language of work or name, welsh)
Question: Is the language of work or name of "brut y brenhinedd" welsh?

Answer the question from context.
Context: Walter is associated with the brut tysilio, which is a welsh chronicle known as brut y brenhinedd.
Question: Is the language of work or name of "brut y brenhinedd" welsh?
Answer: **Yes, "brut y brenhinedd" is a welsh chronicle.**

**Case 2**
Summarize the relations between "elizabeth" and "english" from context.
Context: Annabeth is a female english given name created from a combination of the names anna and elizabeth.
Summarization: The name annabeth is an english female given name that is created from combining the names anna and elizabeth.

Rewrite the triple as a yes/no question.
Triple: (elizabeth, language of work or name, english)
Question: Does elizabeth's language of work or name include english?

Answer the question from context.
Context: The name annabeth is an english female given name that is created from combining the names anna and elizabeth.
Question: Does elizabeth's language of work or name include english?
Answer: **It is not clear from the given context whether elizabeth's language of work or name includes english.**

**Case 3**
Summarize the relations between "luxembourgish wikipedia" and "luxembourgish language" from context.
Context: The luxembourgish wikipedia is the version of wikipedia that is written in the luxembourgish language.
Summarization: The luxembourgish wikipedia is the version of wikipedia that is written in the luxembourgish language.

Rewrite the triple as a yes/no question.
Triple: (luxembourgish wikipedia, language of work or name, luxembourgish language)
Question: Is luxembourgish the language of work or name for the luxembourgish wikipedia?

Answer the question from context.
Context: The luxembourgish wikipedia is the version of wikipedia that is written in the luxembourgish language.
Question: Is luxembourgish the language of work or name for the luxembourgish wikipedia?
Answer: **Luxembourgish is the language of the luxembourgish wikipedia.**

**Case 4**
Summarize the relations between "kuranty" and "russian" from context.
Context: Peter the great replaced the "kuranty" with the first printed newspaper in russian , the "vedomosti".
Summarization: Kuranty was replaced by the first printed newspaper in russian called "vedomosti" by peter the great.

Rewrite the triple as a yes/no question.
Triple: (kuranty, language of work or name, russian)
Question: Is russian the language of work or name for kuranty?

Answer the question from context.
Context: Kuranty was replaced by the first printed newspaper in russian called "vedomosti" by peter the great.
Question: Is russian the language of work or name for kuranty?
Answer: **Russian is the language of the first printed newspaper called "vedomosti", not "kuranty".**

Table 10: Case study regarding relation "language of work or name".

**Case 1**
Summarize the relations between "liṣava river" and "natra river" from context.
Context: The natra river is a tributary of the liṣava river in romania.
Summarization: The natra river is a smaller river that flows into the liṣava river in romania.

Rewrite the triple as a yes/no question.
Triple: (liṣava river, tributary, natra river)
Question: Is the liṣava river a tributary of the natura river?

Answer the question from context.
Context: The natra river is a smaller river that flows into the liṣava river in romania.
Question: Is the liṣava river a tributary of the natura river?
Answer: **No, the natra river flows into the liṣava river, so the liṣava river is not a tributary of the natra river.**

Summarize the relations between "natra river" and "liṣava river" from context.
Context: The natra river is a tributary of the liṣava river in romania.
Summarization: The natra river is a tributary of the liṣava river located in romania.

Rewrite the triple as a yes/no question.
Triple: (natra river, mouth of the watercourse, liṣava river)
Question: Is the mouth of the liṣava river located at the natra river watercourse?

Answer the question from context.
Context: The natra river is a tributary of the liṣava river located in romania.
Question: Is the mouth of the liṣava river located at the natra river watercourse?
Answer: **The context does not provide information to answer this question.**

**Case 2**
Summarize the relations between "danube" and "lech" from context.
Context: His jurisdiction extended at that time from the iller eastward over the lech, north of the danube to the alb, and south to the spurs of the alps.
Summarization: The text mentions that the jurisdiction in question encompassed an area that extended from iller towards east over lech, then further towards north over danube unto alb, and also southwards towards the alps.

Rewrite the triple as a yes/no question.
Triple: (danube, tributary, lech)
Question: Is the lech a tributary of the danube?

Answer the question from context.
Context: The text mentions that the jurisdiction in question encompassed an area that extended from iller towards east over lech, then further towards north over danube unto alb, and also southwards towards the alps.
Question: Is the lech a tributary of the danube?
Answer: **The text implies that the lech is located between the iller to the west and the danube to the north, but it does not explicitly state whether or not the lech is a tributary of the danube.**

Summarize the relations between "blackberry run" and "east branch fishing creek" from context.
Context: the restoration of blackberry run is the third phase of the east branch fishing creek restoration plan.
Summarization: The restoration of blackberry run is part of the third phase of the east branch fishing creek restoration plan.

Rewrite the triple as a yes/no question.
Triple: (blackberry run, mouth of the watercourse, east branch fishing creek)
Question: Does the mouth of the watercourse located in the east branch of the fishing creek have blackberry run?

Answer the question from context.
Context: The restoration of blackberry run is part of the third phase of the east branch fishing creek restoration plan.
Question: Does the mouth of the watercourse located in the east branch of the fishing creek have blackberry run?
Answer: **The context does not provide information about whether the mouth of the watercourse located in the east branch of the fishing creek have blackberry run.**

Table 11: Case study regarding relation "tributary" and "mouth of the watercourse".

| Relation | Template |
| --- | --- |
| per:stateorprovince_of_death | *subject* died in the state or province *object*, Yes or No? |
| per:title | *subject* is a *object*, Yes or No? |
| org:member_of | *subject* is the member of *object*, Yes or No? |
| per:other_family | *subject* is the other family member of *object*, Yes or No? |
| org:country_of_headquarters | *subject* has a headquarter in the country *object*, Yes or No? |
| org:parents | *subject* has the parent company *object*, Yes or No? |
| per:stateorprovince_of_birth | *subject* was born in the state or province *object*, Yes or No? |
| per:spouse | *subject* is the spouse of *object*, Yes or No? |
| per:origin | *subject* has the nationality *object*, Yes or No? |
| per:date_of_birth | *subject* has birthday on *object*, Yes or No? |
| per:schools_attended | *subject* studied in *object*, Yes or No? |
| org:members | *subject* has the member *object*, Yes or No? |
| org:founded | *subject* was founded in *object*, Yes or No? |
| per:stateorprovinces_of_residence | *subject* lives in the state or province *object*, Yes or No? |
| per:date_of_death | *subject* died in the date *object*, Yes or No? |
| org:shareholders | *subject* has shares hold in *object*, Yes or No? |
| org:website | *subject* has the website *object*, Yes or No? |
| org:subsidiaries | *subject* owns *object*, Yes or No? |
| per:charges | *subject* is convicted of *object*, Yes or No? |
| org:dissolved | *subject* dissolved in *object*, Yes or No? |
| org:stateorprovince_of_headquarters | *subject* has a headquarter in the state or province *object*, Yes or No? |
| per:country_of_birth | *subject* was born in the country *object*, Yes or No? |
| per:siblings | *subject* is the siblings of *object*, Yes or No? |
| org:top_members/employees | *subject* has the high level member *object*, Yes or No? |
| per:cause_of_death | *subject* died because of *object*, Yes or No? |
| per:alternate_names | *subject* has the alternate name *object*, Yes or No? |
| org:number_of_employees/members | *subject* has the number of employees *object*, Yes or No? |
| per:cities_of_residence | *subject* lives in the city *object*, Yes or No? |
| org:city_of_headquarters | *subject* has a headquarter in the city *object*, Yes or No? |
| per:children | *subject* is the parent of *object*, Yes or No? |
| per:employee_of | *subject* is the employee of *object*, Yes or No? |
| org:political/religious_affiliation | *subject* has political affiliation with *object*, Yes or No? |
| per:parents | *subject* has the parent *object*, Yes or No? |
| per:city_of_birth | *subject* was born in the city *object*, Yes or No? |
| per:age | *subject* has the age *object*, Yes or No? |
| per:countries_of_residence | *subject* lives in the country *object*, Yes or No? |
| org:alternate_names | *subject* is also known as *object*, Yes or No? |
| per:religion | *subject* has the religion *object*, Yes or No? |
| per:city_of_death | *subject* died in the city *object*, Yes or No? |
| per:country_of_death | *subject* died in the country *object*, Yes or No? |
| org:founded_by | *subject* was founded by *object*, Yes or No? |

Table 12: Templates for TACRED.