# OpenReview forum: "Revisiting Large Language Models as Zero-shot Relation Extractors"
_EMNLP/2023/Conference — EMNLP 2023 Findings_

### Official Review · Reviewer_83on · 2023-08-04

**Soundness:** 3

**Excitement:**

4: Strong: This paper deepens the understanding of some phenomenon or lowers the barriers to an existing research direction.

**Paper Topic And Main Contributions:**

In this paper, the authors propose a new prompting scheme for zero-shot relation extraction. This framework is based on a four step process in which the model is prompted to 1) summarize the context, 2) rewrite the RE triple as a natural question, 3) answer the question based on the simplified context and 4) approximate the probability of each relation uncertainty based on a combination of the probability of each summarized context, generated question and final relation probability. This methodology, termed SUMASK, sharply improves the performance of LLMs over vanilla RE prompting.

**Questions For The Authors:**

-  It seems like the most interesting and strongest motivation for this paper is that it does not require any sort of prompt engineering or template writing to start using. Please correct me if I’m wrong. If this is the case, is there a reason this was not more explicitly discussed in the paper?
   - It is indeed a very important contribution which should be published but proper baselines need to be used.
- Is there a reason for using the uncertainty estimation technique which uses “dispersion degree” for LLMs which do output logits?

**Reasons To Accept:**

- Interesting methodology for leveraging multiple steps of LLM prompting while accounting for the uncertainty in each of the predictions.
- Strong evidence that the current methodology is effective over vanilla prompting in different LLMs.
- Choice of datasets and tasks is thorough.

**Reasons To Reject:**

- The most striking concern with this work is the absence of several relevant baselines from the main experiments even though these baselines are mentioned elsewhere.
  - NLI-DeBERTa and QA4RE are mentioned and compared to in a secondary experiment measuring the NoTA performance but are notably absent from Table 1 without an explanation.
  - NLI-DeBERTa is shown in Table 2 but only as a supervised baseline when it is a zero and few-shot method which could be compared to directly.
  - QA4RE is not mentioned in Table 2 at all even though it is competitive and much simpler than the presented technique. The motivation for not including such baselines is unclear.
-  I was unable to understand Figure 4 from the explanations given, please explain it further.

**Reproducibility:**

3: Could reproduce the results with some difficulty. The settings of parameters are underspecified or subjectively determined; the training/evaluation data are not widely available.

**Reviewer Confidence:**

4: Quite sure. I tried to check the important points carefully. It's unlikely, though conceivable, that I missed something that should affect my ratings.

**Typos Grammar Style And Presentation Improvements:**

- “How does LLMs perform on RE incorporating existing prompt techniques?” L088
- “But it is still unclear whether LLMs are good zero-shot relation extractors by carefully designed prompts. Thus this work aims to investigate the capabilities of LLMs in zero-shot RE” (not sure what this sentence means) L164

---

> ### Author Rebuttal · Authors · 2023-08-29
>
> Thanks for the constructive comments.
>
> ### 1. Absence of several relevant baselines
> Thank you for taking our work so seriously. The reasons for the absence of several relevant baselines from the main experiments are as follows.
>
> - **NLI-DeBERTa and QA4RE are not in Table 1** There are two reasons why we did not summarize the experimental results on FewRel and Wiki-ZSL for these two baselines NLI-DeBERTa and QA4RE. **First**, NLI-DeBERTa is only evaluated in TACRED dataset so we could not directly retrieve the other results from its original paper [1]. And QA4RE [2] follows the NLI-DeBERTa experimental settings. **Second**, both NLI-DeBERTa and QA4RE require the verbalized description of a relation as hypothesis. While the authors of NLI-DeBERTa manually create templates and valid arguments for the 42 relations in TACRED, there are no available crafted verbalized descriptions for FewRel and Wiki-ZSL. And it is a non-trivial task to manually label high-quality and reasonable templates for so many relations. If we use a stiff template like "The relation between *subject* and *object* is *relation*", it actually does not comply with the grammar rules and confuse the relative positional property of the subjects and objects because we have not resolved the specific semantics of the relation label. Therefore, we do not conduct experiments on FewRel and Wiki-ZSL with NLI-DeBERTa and QA4RE. But we would specify the used stiff templates of two baselines and add the re-implementation results in the revision.
>
> - **Zero-shot NLI-DeBERTa is not in Table 2** NLI-DeBERTa is indeed also a zero-shot baseline that can be compared with other methods. We did not put it in Table 2 for two reasons. **First**, the intention of Table 2 is to show the superior performance of SUMASK prompting even compared to the fully supervised methods. Thus the zero-shot methods are not summarized in Table 2. **Second**, we have demonstrated the fully supervised performance of NLI-DeBERTa in Table 2, which is definitely better than zero-shot results. Hence we argue that it is not necessary to put the zero-shot results here compared to many state-of-the-art supervised baselines. Nonetheless, we would add the missing zero-shot results in the Appendix of the revision.
>
> - **QA4RE is not in Table 2** There are several reasons that make the experimental results of QA4RE are not included in Table 2. **First**, the experimental settings of QA4RE and SUMASK is different. QA4RE adopts the micro-F1 as evaluation metrics but does not consider the none-of-the-above (NoTA) relation. This setting is different with other supervised baselines and SUMASK. Thus the results of QA4RE is only evaluated in Table 3 measuring the NoTA-excluded 41-class micro-F1 and NoTA-included 42-class macro F1 on TACRED following [2]. **Second**, the paper of QA4RE is published as a conference paper at ACL 2023 in June. And the source code is uploaded to Github last month. At that time, as the deadline for EMNLP was approaching, we unfortunately did not have enough time to reproduce their method. Now we have obtained the micro-F1 for QA4RE on TACRED using gpt-3.5-turbo and SUMASK surpasses it by around 4.8% absolute F1 score, which indicates that QA4RE is also effective for handling NoTA relation. However, we should note that even though QA4RE is competitive to SUMASK, it requires the verbalized description of relations like NLI-DeBERTa, which is inconvenient and requiring too much human intervention. **Specially**, not all relations can be accurately described by templates because the description of relations may vary across different entities. For instance, the relation *language of work or name* in datast FewRel is hard to describe by a single template, while this relation can not only describe language versions of some literary works, but also describe what language a name belongs to. Consider the following two templates: (1) The language of *subject* is *object*. and (2) *subject* is a name in *object*. The triple (*Elizabeth*, *language of work or name*, *English*) satisfy the second template but deliver confusions in the first template, while the triple (*The Lord of the Rings*, *language of work or name*, *English*) only satisfy the first template. Like you said, SUMASK does not require any sort of prompt engineering or template writing to start using.
>
> We truly appreciate your instructive suggestions that really help we polish the revision.
>
> ### 2. Explanation of Figure 4
> We apologize for the unclear explanations. The Figure 4 is to illustrate that our intuition is consistent with the empirical results. The uncertainty estimation technique is based on an assumption that the outputs of LLMs would be stabler with the predictions equivalent to the ground truth. In other words, if a LLM regards a relation as ground truth in a sentence, then its response to this relation is always "yes" (ie., the uncertainty is low). For the ground truth relations that are not predicted by LLMs, their uncertainty values are relatively high. From Figure 4, it is obviously that most ground truth relations correspond to low uncertainty (< 1), while a fewer correspond to high uncertainty (> 200). Note that Figure 4 indicates that the LLMs consider most gold relations with low uncertainty, making SUMASK has a better chance of choosing ground truth from multiple candidate relations with lowest uncertainty values. Reviewer PmA9 also asks some interesting questions about the uncertainty estimation, you can refer to our replies for better understanding. And we would explain more clearly about this part in the revision.
>
> ### 3. Without prompt engineering or template writing
> Generally, the SUMASK prompting does not require any sort of prompt engineering or template writing to start using. And this is one of our contribution points. Nonetheless, the original purpose of this paper is to show the huge potential of LLMs as zero-shot relation extractors because some recent studies is not optimistic that the performance of prompting LLMs can surpass the performance of fine-tuning small language models. And we design this simple SUMASK prompting without additional human intervention that achieves competitive or superior performance compared to existing zero-shot and fully supervised methods. The decomposition of zero-shot relation extraction and incorporation of uncertainty estimation consistently suggest that proper prompting techniques are beneficial for relation understanding and reasoning.
>
> ### 4. Uncertainty estimation in all LLMs
> The efficiency of using logits to generate the probabilities of intermediate results ([SUMMARIZATION] and [QUESTION]) is relatively low, because we are required to obtain the probability of tokens at each position. Moreover, this method is highly susceptible to extreme values. For example, if a token with a low probability is sampled during the sampling process, it will affect the probability value of the entire sentence. In addition, due to the generated long text sequence, the difference of probability values between generated sentences is not obvious, making it difficult to choose the relation with the highest probability. Using SUMASK with uncertainty estimation brings better outcomes and we show this technique is suitable for both white box (e.g., BLOOM) and black box (e.g., ChatGPT) models.
>
> ### 5. Typos Grammar Style And Presentation Improvements
> Thanks for your suggestions.
> - **“How does LLMs perform on RE incorporating existing prompt techniques?”** The meaning of this sentence is that we are investigating whether some recent techniques such as chain-of-thought or active prompting can bring improvements to LLMs on RE.
> - **“But it is still unclear whether LLMs are good zero-shot relation extractors by carefully designed prompts. Thus this work aims to investigate the capabilities of LLMs in zero-shot RE”** Some previous studies [3][4] claim that LLMs achieve inferior results on RE compared to traditional fine-tuned models. However, they do not design a suitable prompt for RE, but instead adopt the most direct approach like VANILLA prompting. Therefore, we aim to investigate the capabilities of LLMs in zero-shot RE equipping with advanced prompting techniques. And the conclusion is LLMs can become good zero-shot relation extractors with proper prompting techniques.
>
> ### References
>
> [1] Label Verbalization and Entailment for Effective Zero- and Few-Shot Relation Extraction. Sainz et al., 2021. In EMNLP.
>
> [2] Aligning Instruction Tasks Unlocks Large Language Models as Zero-Shot Relation Extractors. Zhang et al., 2023. In Findings of ACL.
>
> [3] Thinking About GPT-3 In-Context Learning for Biomedical IE? Think Again. Jimenez Gutierrez et al., 2022. In Findings of EMNLP.
>
> [4] Large Language Model Is Not a Good Few-shot Information Extractor, but a Good Reranker for Hard Samples! Ma et al., 2023. arXiv preprint arXiv:2303.08559

---

### Official Review · Reviewer_PmA9 · 2023-08-05

**Soundness:** 3

**Excitement:**

3: Ambivalent: It has merits (e.g., it reports state-of-the-art results, the idea is nice), but there are key weaknesses (e.g., it describes incremental work), and it can significantly benefit from another round of revision. However, I won't object to accepting it if my co-reviewers champion it.

**Paper Topic And Main Contributions:**

This paper introduces a novel prompt method to enhance Language Model (LLM) capabilities in solving Relation Extraction (RE) tasks. The authors decompose RE into two subtasks: text summarization and question answering, employing a chain-of-thought process to address the task effectively.

**Questions For The Authors:**

The uncertainty estimation method is indeed fancy and useful. I am curious about the cases with high uncertainty. Figure 4 indicates that many ground truth types have an uncertainty of more than 200. I wonder what level of uncertainty of the other types.

**Reasons To Accept:**

1. The writing of this paper is clear and easy to understand.
2. Their method is effective, achieving competitive zero-shot performance, especially compared to vanilla prompts.
3. The proposed uncertainty estimation method is highly beneficial and contributes significantly to the final success of the model, to some extent.

**Reasons To Reject:**

While they adopt the entity-relation mapping mechanism to handle relation redundancy, my concern remains with the inference complexity of this method. This is due to the fact that it needs to enumerate all possible triples to obtain summarizations, questions, and answers, for k times.

**Reproducibility:**

3: Could reproduce the results with some difficulty. The settings of parameters are underspecified or subjectively determined; the training/evaluation data are not widely available.

**Reviewer Confidence:**

4: Quite sure. I tried to check the important points carefully. It's unlikely, though conceivable, that I missed something that should affect my ratings.

---

> ### Author Rebuttal · Authors · 2023-08-29
>
> Thanks for the constructive comments.
>
> ### 1. Level of uncertainty
> Thank you for your interest in the uncertainty experiments. Figure 4 indicates the correlations between uncertainty estimation and ground truth. The uncertainty values corresponding to different relation samples vary greatly. Specifically, the minimum value of uncertainty is actually 0.0, which indicates that the LLMs are quite confident with its prediction on this ground truth. Take the FewRel as the example, approximately 79.4% of the uncertainty values are between 0 and 1, while 14.7% of the values are between 1 and 200. Only 5.9% of the uncertainty values are greater than 200. Specially, the cases with high uncertainty are due to poor extraction performance of LLMs for certain relations as we discussed in Relation Specific Analysis section and Appendix C.
>
> For other types, the corresponding uncertainty values is mostly relatively high, with only a few being low. Although we make the assumption that LLMs have the relatively low uncertainty regarding to ground truth, other relation types may also corresponding to low uncertainty and cause confusions. Concretely, on all benchmarks, around 82.7% of uncertainty values corresponding to other relation types are greater than 1. There are also 17.3% of uncertainty values are between 0 and 1, indicating LLMs are also confident in treating some other relations as correct labels. Note that the uncertainty values counted here are all based on positive candidate relations (i.e., the LLMs answer with "yes" via the majority vote).
>
> ### 2. Inference complexity
> Typically, the SUMASK prompting method suffers from relatively high inference complexity as it needs to enumerate all possible triples to obtain summarizations, questions, and answers, for $k$ times. Suppose we have $n$ samples to be extracted and $r$ candidate relations, the total complexity is $O(k \times r \times n)$.
> First, using entity types to discard the most irrelevant relations is a useful method, which achieves less complexity $O(k \times \hat{r} \times n)$ where $\hat{r}$ represents the maximum value of the mapped relation candidate set and much less than $r$. Second, we can certainly ask multiple samples (not exceeding the model maximum length) to LLMs at one time to improve inference speed. Suppose we concatenate $k$ samples in a prompt, then the complexity becomes $O(\hat{r} \times n)$.
> More efficient zero-shot prompting for relation extraction is worth exploring in the future. Although the complexity of SUMASK is not advantageous, using SUMASK prompting does not necessitate any form of prompt engineering or template crafting compared to previous work like QA4RE. Thanks for your suggestions.

---

### Official Review · Reviewer_ry3f · 2023-08-10

**Soundness:** 3

**Excitement:**

3: Ambivalent: It has merits (e.g., it reports state-of-the-art results, the idea is nice), but there are key weaknesses (e.g., it describes incremental work), and it can significantly benefit from another round of revision. However, I won't object to accepting it if my co-reviewers champion it.

**Paper Topic And Main Contributions:**

This paper explores the approach of using a large language model (ChatGPT) to address the zero-shot relation extraction problem. The authors introduce a novel prompting technique (Such as Chain of Thought (CoT)) to enhance zero-shot relation extraction, and propose the SUMASK prompting method. This method comprises three steps: Summarization, Question and Answer, and guiding the model for relation extraction through prompts. Through experiments conducted on various benchmark and configuration datasets, the paper thoroughly demonstrates the effectiveness of SUMASK prompting in zero-shot relation extraction.

**Questions For The Authors:**

You can refer to the reviewer's comments, which include feedback on the paper as well as my queries regarding certain aspects of the paper.

**Reasons To Accept:**

1. The introduction of the novel SUMASK prompting method presents a fresh perspective on addressing the issue of zero-shot relation extraction.This approach transforms the relation extraction task into a multi-stage question-answering process, thereby offering a novel avenue for applications in similar downstream tasks.
2.  Through meticulous experimental design and extensive validation, the paper effectively demonstrates the impact of the SUMASK prompting method in the realm of zero-shot relation extraction. This approach exhibits outstanding performance across multiple benchmark datasets, underscoring its practicality and scalability.

**Reasons To Reject:**

While the SUMASK technique proposed by the authors showcases impressive state-of-the-art performance in relation extraction tasks,
the paper does raise two noteworthy concerns that merit further scrutiny：
1.  The ablation experiments presented in Table 5 strongly emphasize the pivotal role of uncertainty estimation methods.Paradoxically, the authors do not place adequate emphasis on explicating the specific contributions stemming from uncertainty estimation—an omission that requires clarification.
2.  As exemplified by the sentence "Savi was born in Pisa, son of Gaetano Savi, professor of Botany at the University of Pisa," it becomes evident that the accurate delineation of intricate relationships among entities—Savi, Pisa, Gaetano Savi, Botany, and University of Pisa—necessitates not just semantic comprehension, but also a degree of logical deduction within the purview of the relation extraction model.Given the discernible efficacy of SUMASK Prompting vis-à-vis VANILLA Prompting, it would be prudent to augment the empirical validation, elucidating how SUMASK Prompting has the potential to enhance the logical reasoning faculties of extensive language models, or to furnish an elucidation of the underlying operational mechanisms intrinsic to SUMASK prompting.

**Reproducibility:**

4: Could mostly reproduce the results, but there may be some variation because of sample variance or minor variations in their interpretation of the protocol or method.

**Reviewer Confidence:**

4: Quite sure. I tried to check the important points carefully. It's unlikely, though conceivable, that I missed something that should affect my ratings.

---

> ### Author Rebuttal · Authors · 2023-08-29
>
> Thanks for the constructive comments.
>
> ### 1. Specific contributions of uncertainty estimation
> The uncertainty estimation method is utilized to improve the reliability of extracted results in black box LLMs such as ChatGPT. This method is important for multi-class classification tasks like relation classification. Because we assume only one valid relation exists in a sentence. And we highlight the substantial role of this method in line 111-114 of the paper **"We further introduce an uncertainty estimation method to approximately characterize output probabilities of LLMs, which yields substantial improvements compared to VANILLA prompting."**
>
> The ablation results demonstrate the pivotal role of uncertainty estimation compared to summarization generation and question generation. Specifically, this technique improves the absolute F1 scores by 19.9% - 36.5% on FewRel, which highlights the importance of uncertainty estimation utilized in relation classification via SUMASK prompting. The specific contributions are not included in the contribution summary, and we would provide some case studies and emphasize this in the revision.
>
>
> ### 2. The empirical validation of SUMASK prompting
>  Thanks for your suggestions. The underlying operational mechanisms intrinsic to SUMASK prompting rely on the strong capabilities of LLMs as zero-shot reasoners [1]. Following the prompt instructions, LLMs pay attention to the entities of interest, infer and summarize the relations between them.
>
> To the best of our knowledge, the logical reasoning faculties of LLMs are not explicitly utilized in previous work of zero-shot relation extraction. From raw text inputs to extracted relation labels, this process lacks intermediate reasoning steps. We decompose the zero-shot relation extraction into three steps to make LLMs sensitize to the semantic understanding and logical reasoning. With the proper instruction for summarization, LLMs are able to perform logical reasoning on specific entities and obtain relations between them. The LLMs can automatically do this via prompting, but the small fine-tuned model cannot.
>
> Therefore, the accurate statement is not that SUMASK "enhances the logical reasoning faculties of extensive language models", but that **"SUMASK elicits the logical reasoning ability inside LLMs for relation deduction"**. The ablation results also show that without the summarization step, the overall performance drops 2.5% - 7.7% F1 on FewRel and TACRED under different settings. The experimental results on overlapping relation extraction also demonstrate the superior of SUMASK prompting over VANILLA prompting.
>
> Here we provide a case study on NYT. The input prompt is:
>
> *Summarize the relations between “Cambodia” and “Penh” from context.*
>
> *Context: Homage to Cambodia was performed at Chaktomuk Conference Hall in Phnom Penh on Oct. 21 , attended by the king.*
>
> *Summarization:*
>
> Then the response from ChatGPT is
>
> *"Phnom Penh" is the capital city of "Cambodia," where the event "Homage to Cambodia" took place at the "Chaktomuk Conference Hall" on October 21.*
>
> Obviously, the first sentence *"Phnom Penh" is the capital city of "Cambodia,"* generated by LLMs clearly elucidates the relationship between two entities, facilitating the subsequent relation prediction including */location/country/administrative_divisions*, */location/location/contains*, */location/administrative_division/country* and */location/country/capital*. We would augment the empirical validation of SUMASK prompting in the revision. Thanks for your instructive suggestions.
>
>
> ### References
> [1] Large Language Models are Zero-shot Reasoners. Kojima et al., 2022. In NeurIPS.

---

### Meta-Review · Area_Chair_M8UT · 2023-09-21

**Recommendation:** 3

**Metareview:**

This paper tackles zero-shot relation extraction and proposes an interesting summarize-and-ask strategy to use LLMs for this challenging task (reviewer 83on gave a good summary of the proposed method). Extensive experiments show fairly strong zero-shot relation extraction performance on multiple standard datasets.

Strengths:
- The main idea is intuitively plausible and may generalizable to other tasks
- The empirical performance, as far as I could tell, is fairly strong for zero-shot RE
- The proposed method doesn't need manually written templates, which is a nice and unique property
- The writing is clear and easy to follow

Weakness
- Added cost for calling LLMs many more times
- There's an inadequate discussion on the uncertainty part, which is quite important. the authors are encouraged to strengthen this part in the revision.

---

### Decision · Program_Chairs · 2023-10-07

**Decision:**

Accept-Findings

**Comment:**

This paper tackles zero-shot relation extraction and proposes an interesting summarize-and-ask strategy to use LLMs for this challenging task (reviewer 83on gave a good summary of the proposed method). Extensive experiments show fairly strong zero-shot relation extraction performance on multiple standard datasets.

Strengths:
- The main idea is intuitively plausible and may generalizable to other tasks
- The empirical performance, as far as I could tell, is fairly strong for zero-shot RE
- The proposed method doesn't need manually written templates, which is a nice and unique property
- The writing is clear and easy to follow

Weakness
- Added cost for calling LLMs many more times
- There's an inadequate discussion on the uncertainty part, which is quite important. the authors are encouraged to strengthen this part in the revision.